# Divide and Couple: Using Monte Carlo Variational Objectives for Posterior Approximation

**Justin Domke**[1] **and Daniel Sheldon**[1,2]
[1] College of Information and Computer Sciences, University of Massachusetts Amherst
[2] Department of Computer Science, Mount Holyoke College

## Abstract

Recent work in variational inference (VI) uses ideas from Monte Carlo estimation to tighten the lower bounds on the log-likelihood that are used as objectives. However, there is no systematic understanding of how optimizing different objectives relates to approximating the posterior distribution. Developing such a connection is important if the ideas are to be applied to inference—i.e., applications that require an approximate posterior and not just an approximation of the log-likelihood. Given a VI objective defined by a Monte Carlo estimator of the likelihood, we use a "divide and couple" procedure to identify augmented proposal and target distributions. The divergence between these is equal to the gap between the VI objective and the log-likelihood. Thus, after maximizing the VI objective, the augmented variational distribution may be used to approximate the posterior distribution.

## 1 Introduction

Variational inference (VI) is a leading approximate inference method in which a posterior distribution $p(z|x)$ is approximated by a simpler distribution $q(z)$ from some approximating family. The procedure to select $q$ is based on the decomposition that [29]

$$\log p(x) = \mathop{\mathbb{E}}_{q(\mathsf{z})} \left[ \log \frac{p(\mathsf{z}, x)}{q(\mathsf{z})} \right] + \mathrm{KL}[q(\mathsf{z})\|p(\mathsf{z}|x)]. \tag{1}$$

The first term is the *evidence lower bound* (ELBO) [4]. Selecting $q$ to maximize the ELBO tightens the lower bound on $\log p(x)$ and simultaneously minimizes the KL-divergence in the second term. This dual view is important because minimizing the KL-divergence justifies using $q$ to approximate the posterior for making predictions.

Recent work has investigated tighter objectives [6, 20, 18, 17, 23, 25], based on the following principle: Let $R$ be an estimator of the likelihood—i.e., a nonnegative random variable with $\mathbb{E}\,R = p(x)$. By Jensen's inequality, $\log p(x) \geq \mathbb{E}\log R$, so $\log R$ is a stochastic lower bound on the log-likelihood. Parameters of the estimator can be optimized to tighten the bound. Standard VI is the case when $R = p(\mathsf{z}, x)/q(\mathsf{z})$ and $\mathsf{z} \sim q$, which is parameterized in terms of $q$. Importance-weighted autoencoders [IWAEs; 6] essentially use $R = \frac{1}{M}\sum_{m=1}^{M} p(\mathsf{z}_m, x)/q(\mathsf{z}_m)$ where $\mathsf{z}_1 \ldots \mathsf{z}_M \sim q$ are iid. Sequential Monte Carlo (SMC) also gives a variational objective [20, 18, 17]. The principle underlying these works is that likelihood estimators that are more concentrated lead to tighter bounds, because the gap in Jensen's inequality is smaller. To date, the main application has been for learning parameters of the generative model $p$.

Our key question is: *what are the implications of modified variational objectives for probabilistic inference*? Eq. (1) relates the standard ELBO to $\mathrm{KL}[q\|p]$, which justifies using $q$ for posterior inference. If we optimize a variational objective obtained from a different estimator, does this still correspond to minimizing some KL-divergence? It has been shown [10, 20, 11] that maximizing the IWAE objective corresponds to minimizing (an upper bound to) $\mathrm{KL}[q^{\mathrm{IS}}(\mathsf{z})\|p(\mathsf{z}|x)]$, where $q^{\mathrm{IS}}$ is a version of $q$ that is

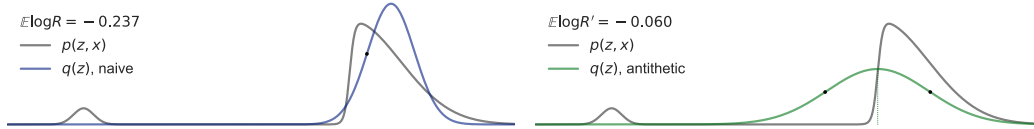

Figure 1: Left: Naive VI on the running example. Right: A tighter bound using antithetic sampling.

"corrected" toward $p$ using importance sampling; this justifies using $q^{\text{IS}}$ to approximate the posterior. Naesseth et al. [20] also show that performing VI with an SMC objective can be seen as minimizing (an upper bound to) a divergence from the SMC sampling distribution $q^{\text{SMC}}(z)$ to $p(z|x)$. For an arbitrary estimator, however, there is little understanding.

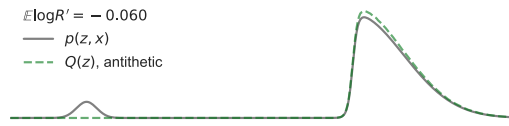

Figure 2: A kernel density approximation of $Q(z)$ for the antithetic estimator in Fig. 1.

We establish a deeper connection between variational objectives and approximating families. Given a non-negative Monte Carlo (MC) estimator $R$ such that $\mathbb{E} R = p(x)$, we show how to find a distribution $Q(z)$ such that the divergence between $Q(z)$ and $p(z|x)$ is at most the gap between $\mathbb{E} \log R$ and $\log p(x)$. Thus, better estimators mean better posterior approximations. The approximate posterior $Q(z)$ can be found by a two-step "divide and couple" procedure. The "divide" step follows Le et al. [17] and connects maximizing $\mathbb{E} \log R$ to minimizing a divergence between two distributions, but not necessarily involving $p(z|x)$. The "couple" step shows how to find $Q(z)$ such that the divergence is an upper bound to $\text{KL}\left[Q(\mathsf{z})\|p(\mathsf{z}|x)\right]$. We show how a range of ideas from the statistical literature—such as antithetic sampling, stratified sampling, and quasi Monte Carlo [24]—can produce novel variational objectives; then, using the divide and couple framework, we describe efficient-to-sample approximate posteriors $Q(z)$ for each of these objectives. We contribute mathematical tools for deriving new estimators and approximating distributions within this framework. Experiments show that the novel objectives enabled by this framework can lead to improved likelihood bounds and posterior approximations.

There is a large body of work using MC techniques to reduce the variance of *gradient estimators* of the standard variational objective [26, 5, 19, 31, 13]. The aims of this paper are different: we use MC techniques to change $R$ to get a tighter objective.

## 2   Setup and Motivation

Imagine we have some distribution $p(z, x)$. After observing data $x$, we wish to approximate the posterior $p(z|x)$. Traditional VI tries to both bound $\log p(x)$ and approximate $p(z|x)$ using the "ELBO decomposition" of Eq. (1). We already observed that a similar lower bound can be obtained from any non-negative random variable $R$ with $\mathbb{E} R = p(x)$, since by Jensen's inequality,

$$\log p(x) \geq \mathbb{E} \log R.$$

Traditional VI can be seen as defining $R = p(\mathsf{z}, x)/q(\mathsf{z})$ for $\mathsf{z} \sim q$ and then optimizing the parameters of $q$ to maximize $\mathbb{E} \log R$. Many other estimators $R$ of $p(x)$ can be designed and their parameters optimized to make $\mathbb{E} \log R$ as large as possible. We want to know: what relationship does this have to approximating the posterior $p(z|x)$?

### 2.1   Example

Fig. 1 shows a one dimensional target distribution $p(z, x)$ as a function of $z$, and the Gaussian $q(z)$ obtained by standard VI, i.e. maximizing $\mathbb{E} \log R$ for $R = p(\mathsf{z}, x)/q(\mathsf{z})$. The resulting bound is $\mathbb{E} \log R \approx -0.237$, while the true value is $\log p(x) = 0$. By Eq. (1), the KL-divergence from $q(z)$ to $p(z|x)$ is $0.237$. Tightening the likelihood bound has made $q$ close to $p$.

A Gaussian cannot exactly represent the main mode of $p(z, x)$, since it is asymmetric. Antithetic sampling [24] can exploit this. Define

$$R' = \frac{1}{2}\left(\frac{p(\mathsf{z}, x) + p(T(\mathsf{z}), x)}{q(\mathsf{z})}\right), \; \mathsf{z} \sim q, \tag{2}$$

where $T(z) = \mu - (z - \mu)$ is $z$ "reflected" around the mean $\mu$ of $q$. This is a valid estimator since $q(z)$ is constant under reflection. Tightening this bound over Gaussians $q$ gives $\mathbb{E} \log R' \approx -0.060$. This is better, intuitively, since the right half of $q$ is a good match to the main mode of $p$, i.e. since $q(z) \approx \frac{1}{2}p(z, x)$ for $z$ in that region.

What about $p(z|x)$? It is *not* true that antithetic sampling gives a $q(z)$ with lower divergence (it is around 7.34). After all, naive VI already found the optimal Gaussian. Is there some *other* distribution that is close to $p(z|x)$? How can we find it? These questions motivate this paper.

## 2.2 Notation and Conventions

We use sans-serif font for random variables. All estimators $R$ may depend on the input $x$, but we suppress this for simplicity. Similarly, $a(z|\omega)$ may depend on $x$. Proofs for all results are in the supplement. Objects such as $P^{\mathrm{MC}}$, $Q$, $p$, $q$ are distributions of random variables and will be written like densities: $Q(\omega)$, $p(z, x)$. However, the results are more general: the supplement includes a more rigorous version of our main results using probability measures. We write densities with Dirac delta functions. These are not Lebesgue-integrable, but can be interpreted unambiguosuly as Dirac measures: e.g., $a(z|\omega) = \delta(z - \omega)$ means the conditional distribution of z given $\omega = \omega$ is the Dirac measure $\delta_\omega$. Throughout, $x$ is fixed, so $p(z, x)$ and $P^{\mathrm{MC}}(\omega, z, x)$ are unnormalized distributions over the other variables, and $p(z|x)$ and $P^{\mathrm{MC}}(\omega, z|x)$ are the corresponding normalized distributions.

## 3 The Divide-and-Couple Framework

In this section we identify a correspondence between maximizing a likelihood bound and posterior inference for general non-negative estimators using a two step "divide" and then "couple" construction.

### 3.1 Divide

Let $R(\boldsymbol{\omega})$ be a positive function of $\boldsymbol{\omega} \sim Q(\omega)$ such that $\mathbb{E}_{Q(\boldsymbol{\omega})} R(\boldsymbol{\omega}) = p(x)$, i.e., $R$ is an unbiased likelihood estimator with sampling distribution $Q(\omega)$. The "divide" step follows [17, Claim 1]: we can interpret $\mathbb{E}_{Q(\boldsymbol{\omega})} \log R(\boldsymbol{\omega})$ as an ELBO by defining $P^{\mathrm{MC}}$ so that $R(\omega) = P^{\mathrm{MC}}(\omega, x)/Q(\omega)$. That is, $P^{\mathrm{MC}}$ and $Q$ "divide" to produce $R$. Specifically:

**Lemma 1.** *Let $\boldsymbol{\omega}$ be a random variable with distribution $Q(\omega)$ and let $R(\boldsymbol{\omega})$ be a positive estimator such that $\mathbb{E}_{Q(\boldsymbol{\omega})} R(\boldsymbol{\omega}) = p(x)$. Then*

$$P^{MC}(\omega, x) \quad = \quad Q(\omega)R(\omega)$$

*is an unnormalized distribution over $\omega$ with normalization constant $p(x)$ and $R(\omega) = P^{MC}(\omega, x)/Q(\omega)$ for $Q(\omega) > 0$. Furthermore as defined above,*

$$\log p(x) = \mathop{\mathbb{E}}_{Q(\boldsymbol{\omega})} \log R(\boldsymbol{\omega}) + \mathrm{KL}\left[Q(\boldsymbol{\omega}) \middle\| P^{MC}(\boldsymbol{\omega}|x)\right]. \tag{3}$$

While this shows that it is easy to connect a stochastic likelihood bound to minimizing a KL-divergence, this construction alone is not useful for probabilistic inference, since neither $Q(\omega)$ nor $P^{\mathrm{MC}}(\omega, x)$ make any reference to $z$. Put another way: Even if $\mathrm{KL}\left[Q(\boldsymbol{\omega}) \middle\| P^{\mathrm{MC}}(\boldsymbol{\omega}|x)\right]$ is small, so what? This motivates the coupling step below. More generally, $P^{\mathrm{MC}}$ is defined by letting $R = dP^{\mathrm{MC}}/dQ$ be the Radon-Nikodym derivative, a change of measure from $Q$ to $P^{\mathrm{MC}}$; see supplement.

### 3.2 Couple

If the distributions identified in the above lemma are going to be useful for approximating $p(z|x)$, they must be connected somehow to $z$. In this section, we suggest *coupling* $P^{\mathrm{MC}}(\omega, x)$ and $p(z, x)$ into some new distribution $P^{\mathrm{MC}}(\omega, z, x)$ with $P^{\mathrm{MC}}(z, x) = p(z, x)$. In practice, it is convenient to describe couplings via a conditional distribution $a(z|\omega)$ that augments $P^{\mathrm{MC}}(\omega, x)$. There is a straightforward condition for when $a$ is valid: for the augmented distribution $P^{\mathrm{MC}}(z, \omega, x) = P^{\mathrm{MC}}(\omega, x)a(z|\omega)$ to be a valid coupling, we require that $\int P^{\mathrm{MC}}(\omega, x)a(z|\omega)d\omega = p(z, x)$. An equivalent statement of this requirement is as follows.

**Definition.** An estimator $R(\omega)$ and a distribution $a(z|\omega)$ are a *valid estimator-coupling pair* under distribution $Q(\omega)$ if

$$\mathbb{E}_{Q(\boldsymbol{\omega})} R(\boldsymbol{\omega})a(z|\boldsymbol{\omega}) = p(z,x). \tag{4}$$

The definition implies that $\mathbb{E}_{Q(\boldsymbol{\omega})} R(\boldsymbol{\omega}) = p(x)$ as may be seen by integrating out $z$ from both sides. That is, the definition implies that $R$ is an unbiased estimator for $p(x)$.

Now, suppose we have a valid estimator/coupling pair and that $R$ is a good (low variance) estimator. How does this help us to approximate the posterior $p(z|x)$? The following theorem gives the "divide and couple" framework.[1]

**Theorem 2.** Suppose that $R(\omega)$ and $a(z|\omega)$ are a valid estimator-coupling pair under $Q(\omega)$. Then,

$$Q(z,\omega) = Q(\omega)a(z|\omega), \tag{5}$$
$$P^{\mathrm{MC}}(z,\omega,x) = Q(\omega)R(\omega)a(z|\omega), \tag{6}$$

are valid distributions, $P^{\mathrm{MC}}(z,x) = p(z,x)$, and

$$\log p(x) = \mathop{\mathbb{E}}_{Q(\boldsymbol{\omega})} \log R(\boldsymbol{\omega}) + \mathrm{KL}\left[Q(\mathsf{z})\|p(\mathsf{z}|x)\right] + \mathrm{KL}\left[Q(\boldsymbol{\omega}|\mathsf{z})\big\|P^{\mathrm{MC}}(\boldsymbol{\omega}|\mathsf{z},x)\right]. \tag{7}$$

The final term in Eq. (7) is a conditional divergence [9, Sec. 2.5]. This is the divergence over $\omega$ between $Q(\omega|z)$ and $P^{\mathrm{MC}}(\omega|z,x)$, averaged over $z$ drawn from $Q(z)$. At a high level, this theorem is proved by applying Lem. 1, using the chain rule of KL-divergence, and simplifying using the fact that $a$ is a valid coupling. The point of this theorem is: if $R$ is a good estimator, then $\mathbb{E}\log R$ will be close to $\log p(x)$. Since KL-divergences are non-negative, this means that the marginal $Q(z)$ must be close to the target $p(z|x)$. A coupling gives us a way to *transform* $Q(\omega)$ so as to approximate $p(z|x)$.

To be useful, we need to access $Q(z)$, usually by sampling. We assume it is possible to sample from $Q(\omega)$ since this is part of the estimator. The user must supply a routine to sample from $a(z|\omega)$. This is why the "trivial coupling" of $a(z|\omega) = p(z|x)$ is not helpful — if the user could sample from $p(z|x)$, the inference problem is already solved! Some estimators may be pre-equipped with a method to approximate $p(z|x)$. For example, in SMC, $\omega$ include particles $z_i$ and weights $w_i$ such that selecting a particle in proportion to its weight approximates $p(z|x)$. This can be seen as a coupling $a(z|\omega)$, which provides an alternate interpretation of the divergence bounds of [20]. The divide and couple framework is also closely related to extended-space MCMC methods [2], which also use extended target distributions that admit $p(z,x)$ as a marginal; see also Sec. 6. However, in these methods, the estimators also seem to come with "obvious" couplings. There is no systematic understanding of how to derive couplings for general estimators.

### 3.3 Example

Consider again the antithetic estimator from Eq. (2). We saw before that the antithetic estimator gives a tighter variational bound than naive VI under the distribution in Fig. 1. However, that distribution is *less* similar to the target than the one from naive VI. To reflect this (and match our general notation) we now[2] write $\boldsymbol{\omega} \sim Q$ (instead of $\mathsf{z} \sim q$).

Now, we again ask: Since $Q(\omega)$ is a poor approximation of the target, can we find some other distribution that is a good approximation? Consider the coupling distribution

$$a(z|\omega) = \pi(\omega)\,\delta(z-\omega) + (1-\pi(\omega))\,\delta(z-T(\omega)), \quad \pi(\omega) = \frac{p(\omega,x)}{p(\omega,x)+p(T(\omega),x)}.$$

Intuitively, $a(z|\omega)$ is supported only on $z=\omega$ and on $z=T(\omega)$, with probability proportional to the target distribution. It is simple to verify (by substitution, see Claim 5 in supplement) that $R$ and $a$ form a valid estimator-coupling pair. Thus, the augmented variational distribution is $Q(\omega,z) = Q(\omega)a(z|\omega)$. To sample from this, draw $\omega \sim Q$ and select $z=\omega$ with probability $\pi(\omega)$, or $z=T(\omega)$ otherwise. The marginal $Q(z)$ is shown in Fig. 2. This is a much better approximation of the target than naive VI.

Table 1: Variance reduction methods jointly transform estimators and couplings. Take an estimator $R_0(\omega)$ with coupling $a_0(z|\omega)$, valid under $Q_0(\omega)$. Each line shows a new estimator $R(\cdot)$ and coupling $a(z|\cdot)$. The method to simulate $Q(\cdot)$ is described in the left column. Here, $F^{-1}$ is a mapping so that if $\omega$ is uniform on $[0,1]^d$, then $F^{-1}(\omega)$ has density $Q_0(\omega)$.

| Description | $R(\cdot)$ | $a(z\|\cdot)$ |
|---|---|---|
| **IID Mean** <br> $\omega_1 \cdots \omega_M \sim Q_0$ i.i.d. | $\dfrac{1}{M}\displaystyle\sum_{m=1}^{M} R_0(\omega_m)$ | $\dfrac{\sum_{m=1}^{M} R_0(\omega_m) a_0(z\|\omega_m)}{\sum_{m=1}^{M} R_0(\omega_m)}$ |
| **Stratified Sampling** <br> $\Omega_1 \cdots \Omega_M$ partition $\Omega$, <br> $\omega_1^m \cdots \omega_{M_n}^1 \sim Q_0$ restricted to $\Omega_m$, <br> $\mu_m = Q_0(\omega \in \Omega_m)$. | $\displaystyle\sum_{m=1}^{M} \dfrac{\mu_m}{N_m} \sum_{n=1}^{N_m} R_0\left(\omega_n^m\right)$ | $\dfrac{\sum_{m=1}^{M} \frac{\mu_m}{N_m} \sum_{n=1}^{N_m} R_0\left(\omega_n^m\right) a_0(z\|\omega_n^m)}{\sum_{m=1}^{M} \frac{\mu_m}{N_m} \sum_{n=1}^{N_m} R_0\left(\omega_n^m\right)}$ |
| **Antithetic Sampling** <br> $\omega \sim Q_0$. For all $m$, $T_m(\omega) \overset{d}{=} \omega$. | $\dfrac{1}{M}\displaystyle\sum_{m=1}^{M} R_0\left(T_m(\omega)\right)$ | $\dfrac{\sum_{m=1}^{M} R_0(T_m(\omega)) \, a_0\left(z\|T_m(\omega)\right)}{\sum_{m=1}^{M} R(T_m(\omega))}$ |
| **Randomized Quasi Monte Carlo** <br> $\omega \sim \text{Unif}([0,1]^d)$, $\bar{\omega}_1, \cdots \bar{\omega}_M$ fixed, <br> $T_m(\omega) = F^{-1}\left(\bar{\omega}_m + \omega \pmod 1\right)$ | $\dfrac{1}{M}\displaystyle\sum_{m=1}^{M} R_0\left(T_m(\omega)\right)$ | $\dfrac{\sum_{m=1}^{M} R_0(T_m(\omega)) \, a_0\left(z\|T_m(\omega)\right)}{\sum_{m=1}^{M} R_0(T_m(\omega))}$ |
| **Latin Hypercube Sampling** <br> $\omega_1, \cdots, \omega_M$ jointly sampled from Latin hypercube [24, Ch. 10.3], $T = F^{-1}$. | $\dfrac{1}{M}\displaystyle\sum_{m=1}^{M} R_0\left(T(\omega_m)\right)$ | $\dfrac{\sum_{m=1}^{M} R_0(T(\omega_m)) \, a_0\left(z\|T(\omega_m)\right)}{\sum_{m=1}^{M} R_0(T(\omega_m))}$ |

## 4 Deriving Couplings

Thm. 2 says that if $\mathbb{E}_{Q(\omega)} \log R(\omega)$ is close to $\log p(x)$ and you have a tractable coupling $a(z|\omega)$, then drawing $\omega \sim Q(\omega)$ and then $z \sim a(z|\omega)$ yields samples from a distribution $Q(z)$ close to $p(z|x)$. But how can we find a tractable coupling?

Monte Carlo estimators are often created recursively using techniques that take some valid estimator $R$ and transform it into a new valid estimator $R'$. These techniques (e.g. change of measure, Rao-Blackwellization, stratified sampling) are intended to reduce variance. Part of the power of Monte Carlo methods is that these techniques can be easily combined. In this section, we extend some of these techniques to transform valid *estimator-coupling pairs* into new valid estimator-coupling pairs. The hope is that the standard toolbox of variance reduction techniques can be applied as usual, and the coupling is derived "automatically".

Table 1 shows corresponding transformations of estimators and couplings for several standard variance reduction techniques. In the rest of this section, we will give two abstract tools that can be used to create all the entries in this table. For concreteness, we begin with a trivial "base" estimator-coupling pair. Take a distribution $Q_0(\omega)$ and let $R_0(\omega) = p(\omega, x)/Q_0(\omega)$ and $a_0(z|\omega) = \delta(z - \omega)$ (the deterministic coupling). It is easy to check that these satisfy Eq. (4).

### 4.1 Abstract Transformations of Estimators and Couplings

Our first abstract tool transforms an estimator-coupling pair on some space $\Omega$ into another estimator-coupling pair on a space $\Omega^M \times \{1, \cdots, M\}$. This can be thought of as having $M$ "replicates" of the $\omega$ in the original estimator, along with an extra integer-valued variable that selects one of them. We emphasize that this result does not (by itself) reduce variance — in fact, $R$ has exactly the same distribution as $R_0$.

**Theorem 3.** Suppose that $R_0(\omega)$ and $a_0(z|\omega)$ are a valid estimator-coupling pair under $Q_0(\omega)$. Let $Q(\omega_1, \cdots, w_M, m)$ be any distribution such that if $(\omega_1, \cdots, \omega_M, \mathsf{m}) \sim Q$, then $\omega_{\mathsf{m}} \sim Q_0$. Then,

$$R(\omega_1, \cdots, \omega_M, m) = R_0(\omega_m) \tag{8}$$

$$a(z|\omega_1, \cdots, \omega_M, m) = a_0(z|\omega_m) \tag{9}$$

are a valid estimator-coupling pair under $Q(\omega_1, \cdots, w_M, m)$.

Rao-Blackwellization is a well-known way to transform an estimator to reduce variance; we want to know how it affects *couplings*. Take an estimator $R_0(\omega, \nu)$ with state space $\Omega \times N$ and distribution

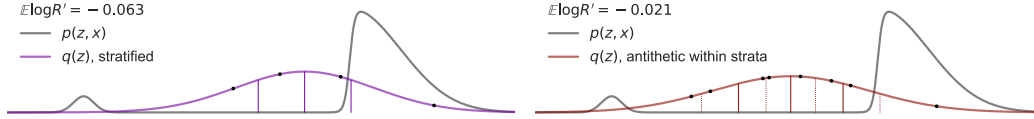

Figure 3: Stratified sampling and antithetic within stratified sampling on the running example.

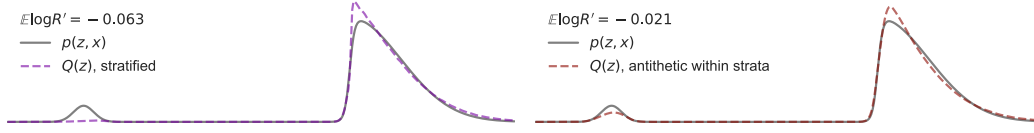

Figure 4: A kernel density approximation of $Q(z|x)$ for the estimators in Fig. 3.

$Q_0(\omega, \nu)$. A new estimator $R(\omega) = \mathbb{E}_{Q_0(\mathbf{v}|\omega)} R(\omega, \mathbf{v})$ that analytically marginalizes out $\nu$ has the same expectation and equal or lesser variance, by the Rao-Blackwell theorem. The following result shows that if $R_0$ had a coupling, then it is easy to define a new coupling for $R$.

**Theorem 4.** Suppose that $R_0(\omega, \nu)$ and $a_0(z|\omega, \nu)$ are a valid estimator-coupling pair under $Q_0(\omega, \nu)$. Then

$$R(\omega) = \mathop{\mathbb{E}}_{Q_0(\mathbf{v}|\omega)} R_0(\omega, \mathbf{v}),$$

$$a(z|\omega) = \frac{1}{R(\omega)} \mathop{\mathbb{E}}_{Q_0(\mathbf{v}|\omega)} \left[ R_0(\omega, \mathbf{v}) \, a_0(z|\omega, \mathbf{v}) \right],$$

are a valid estimator-coupling pair under $Q(\omega) = \int Q_0(\omega, \nu) d\nu$.

### 4.2 Specific Variance Reduction Techniques

Each of the techniques in Table 1 can be derived by first applying Thm. 3 and then Thm. 4. As a simple example, consider the IID mean. Suppose $R_0(\omega)$ and $a_0(z|\omega)$ are valid under $Q_0$. If we let $\boldsymbol{\omega}_1, \cdots, \boldsymbol{\omega}_M \sim Q_0$ i.i.d. and $\mathsf{m}$ uniform on $\{1, \cdots, M\}$ then this satisfies the condition of Thm. 3 that $\boldsymbol{\omega}_\mathsf{m} \sim Q_0$. Thus we can define $R$ and $a$ as in Eq. (8) and Eq. (9). Applying Rao-Blackwellization to marginalize out $\mathsf{m}$ gives exactly the form for $R$ and $a$ shown in the table. Details are in Sec. 9 of the supplement

As another example, take stratified sampling. For simplicity, we assume one sample in each strata ($N_m = 1$). Suppose $\Omega_1 \cdots \Omega_M$ partition the state-space and let $\boldsymbol{\omega}_m \sim Q_0(\boldsymbol{\omega}|\boldsymbol{\omega} \in \Omega_m)$ and $\mathsf{m}$ be equal to $m$ with probability $\mu_m = Q_0(\omega \in \Omega_m)$. It is again the case that $\boldsymbol{\omega}_\mathsf{m} \sim Q_0$, and applying Thm. 3 and then Thm. 4 produces the estimator-coupling pair shown in the table. Again, details are in Sec. 9 of the supplement.

### 4.3 Example

We return to the example from Sec. 2.1 and Sec. 3.3. Fig. 3 shows the result of applying stratified sampling to the standard VI estimator $R = p(\mathsf{z}, x)/q(\mathsf{z})$ and then adjusting the parameters of $q$ to tighten the bound. The bound is tighter than standard VI and slightly worse than antithetic sampling.

Why not combine antithetic and stratified sampling? Fig. 3 shows the result of applying antithetic sampling inside of stratified sampling. Specifically, the estimator $R(\omega^m)$ for each stratum $m$ is replaced by $\frac{1}{2}\left(R(\omega^m) + R(T_m(\omega^m))\right)$ where $T_m$ is a reflection inside the stratum that leaves the density invariant. A fairly tight bound results. For all of antithetic sampling (Fig. 2), stratified sampling (Fig. 3) and antithetic within stratified sampling (Fig. 3) tightening $\mathbb{E} \log R$ finds $Q(\omega)$ such that all batches place some density on $z$ in the main mode of $p$. Thus, the better sampling methods permit a $q$ with some coverage of the left mode of $p$ while precluding the possibility that all samples in a batch are simultaneously in a low-density region (which would result in $R$ near zero, and thus a very low value for $\mathbb{E} \log R$). What do these estimators say about $p(z|x)$? Fig. 4 compares the resulting $Q(z)$ for each estimator — the similarity to $p(z|x)$ correlates with the likelihood bound.

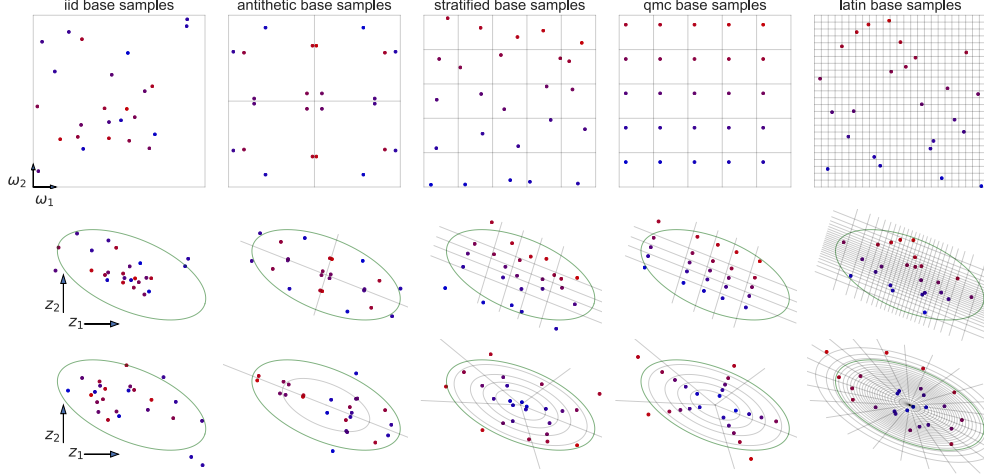

Figure 5: Different sampling methods applied to Gaussian VI. Top row: Different methods to sample from the unit cube. Middle row: these samples transformed using the "Cartesian" mapping. Bottom row: Same samples transformed using the "Elliptical" mapping.

## 5 Implementation and Empirical Study

Our results are easy to put into practice, e.g. for variational inference with Gaussian approximating distributions and the reparameterization trick to estimate gradients.. To illustrate this, we show a simple but general approach. As shown in Fig. 5 the idea is to start with a batch of samples $\omega_1 \cdots \omega_M$ generated from the unit hypercube. Different sampling strategies can give more uniform coverage of the cube than i.i.d. sampling. After transformation, one obtains samples $z_1 \cdots z_M$ that have more uniform coverage of the Gaussian. This better coverage often manifests as a lower-variance estimator $R$. Our coupling framework gives a corresponding approximate posterior $Q(z)$.

Formally, take any distribution $Q(\omega_1, \cdots, \omega_M)$ such that each marginal $Q(\omega_m)$ is uniform over the unit cube (but the different $\omega_m$ may be dependent). As shown in Fig. 5, there are various ways to generate $\omega_1 \cdots \omega_M$ and to map them to samples $z_1 \cdots z_M$ from a Gaussian $q(z_m)$. Then, Fig. 6 gives algorithms to generate an estimator $R$ and to generate $z$ from a distribution $Q(z)$ corresponding to a valid coupling. We use mappings $\omega \overset{F^{-1}}{\to} u \overset{\mathcal{T}_\theta}{\to} z$ where $t_\theta = \mathcal{T}_\theta \circ F^{-1}$ maps $\boldsymbol{\omega} \sim \mathrm{Unif}([0,1]^d)$ to $t_\theta(\boldsymbol{\omega}) \sim q_\theta$ for some density $q_\theta$. The idea is to implement variance reduction to sample (batches of) $\omega$, use $F^{-1}$ to map $\omega$ to a "standard" distribution (typically in the same family as $q_\theta$), and then use $\mathcal{T}_\theta$ to map samples from the standard distribution to samples from $q_\theta$.

The algorithms are again derived from Thm. 3 and Thm. 4. Define $Q_0(\omega)$ uniform on $[0,1]^d$, $R_0(\omega) = p(t_\theta(\omega), x)/q_\theta(t_\theta(\omega))$ and $a_0(z|\omega) = \delta(z - t_\theta(\omega))$. These define a valid estimator-coupling pair. Let $Q(\omega_1, \cdots, \omega_M)$ be as described (uniform marginals) and $\mathsf{m}$ uniform on $\{1, \cdots, M\}$. Then $Q(\omega_1, \cdots, \omega_M, \mathsf{m})$ satisfies the assumptions of Thm. 3, so we can use that theorem then Thm. 4 to Rao-Blackwellize out $\mathsf{m}$. This produces the estimator-coupling pair in Fig. 6.

**Algorithm** (Generate $R$)

- Generate $\omega_1, \cdots, \omega_M$ from any distribution where $\omega_m$ is marginally uniform over $[0,1]^d$.
- Map to a standard dist. as $u_m = F^{-1}(\omega_m)$.
- Map to $q_\theta$ as $z_m = \mathcal{T}_\theta(u_m)$.
- Return $R = \frac{1}{M} \sum_{m=1}^M \frac{p(z_m, x)}{q_\theta(z_m)}$

**Algorithm** (Sample from $Q(z)$)

- Generate $z_1, \cdots z_M$ as on the left.
- For all $m$ compute weight $w_m = \frac{p(z_m, x)}{q_\theta(z_m)}$.
- Select $m$ with probability $\frac{w_m}{\sum_{m'=1}^M w_{m'}}$.
- Return $z_m$

Figure 6: Generic methods to sample $R$ (left) and $Q(z)$ (right). Here, $Q(\omega_1, \cdots, \omega_M)$ is any distribution where the marginals $Q(\omega_m)$ are uniform over the unit hypercube.

The value of this approach is the many off-the-shelf methods to generate "batches" of samples $(\omega_1, \cdots, \omega_M)$ that have good "coverage" of the unit cube. This manifests as coverage of $q_\theta$ after being mapped. Fig. 5 shows examples of this with multivariate Gaussians. As shown, there may be multiple mappings $F^{-1}$. These manifest as different coverage of $q_\theta$, so the choice of mapping influences the quality of the estimator. We consider two examples: The "Cartesian" mapping $F_\mathcal{N}^{-1}(\omega)$ simply applies the inverse CDF of the standard Gaussian. An "elliptical" mapping, meanwhile, uses the "elliptical" reparameterization of the Gaussian [11]: If $r \sim \chi_d$ and $v$ is uniform over the unit sphere, then $r\, v \sim \mathcal{N}(0, I)$. In Fig. 5 we generate $r$ and $v$ from the uniform distribution as $r = F_{\chi_d}^{-1}(\omega_1)$ and $v = (\cos(2\pi\omega_2), \sin(2\pi\omega_2))$, and then set $F^{-1}(\omega) = r\, v$. In higher dimensions, it is easier to generate samples from the unit sphere using redundant dimensions. Thus, we use $\omega \in \mathbb{R}^{d+1}$ and map the first component to $r$ again using the inverse $\chi$ distribution CDF $F_{\chi_d}^{-1}$. The other components are mapped to the unit sphere by first applying the Gaussian inverse CDF in each component, then normalizing.

In the experiments, we use a multivariate Gaussian $q_\theta$ with parameters $\theta = (C, \mu)$. The mapping is $\mathcal{T}_\theta(u) = Cu + \mu$. To ensure a diverse test, we downloaded the corpus of models from the Stan [7] model library [30] (see also Regier et al. [27]) and created an interface for automatic differentiation in Stan to interoperate with automatic differentiation code written in Python. We compare VI in terms of the likelihood bound and in terms of the (squared Frobenius norm) error in the estimated posterior variance. As a surrogate for the true variance, we computed the empirical variance of 100,000 samples generated via Stan's Hamiltonian Markov chain Monte Carlo (MCMC) method. For tractability, we restrict to the 88 models where profiling indicates MCMC would take at most 10 hours, and evaluating the posterior for 10,000 settings of the latent variables would take at most 2 seconds. It was infeasible to tune stochastic gradient methods for all models. Instead we used a fixed batch of 50,000 batches $\omega_1, \cdots, \omega_M$ and optimized the empirical ELBO using BFGS, initialized using Laplace's method. A fresh batch of 500,000 samples was used to compute the final likelihood bound and covariance estimator. Fig. 7 shows example errors for a few models. The supplement contains similar plots for all models, as well as plots aggregating statistics, and a visualization of how the posterior density approximation changes.

## 6  Conclusions

Recent work has studied the use of improved Monte Carlo estimators for better variational likelihood bounds. The central insight of this paper is that an *approximate posterior* can be constructed from an estimator using a *coupling*. This posterior's divergence is bounded by the looseness of the likelihood bound. We suggest a framework of "estimator-coupling" pairs to make this coupling easy to construct for many estimators.

Several recent works have viewed Monte Carlo VI bounds through the lens of augmented VI [3, 10, 20, 11, 16]. These establish connections between particular likelihood estimators and approximate posteriors through extended distributions. They differ from our work primarily in the "direction" of the construction, the generality, or both. Most of the work uses the following reasoning, which starts with an approximate posterior and arrives at a tractable likelihood estimator. Take a Monte Carlo method (e.g. self-normalized importance sampling) to approximately sample from $p(z|x)$. Call the approximation $q(z)$, but suppose it is not tractable to evaluate $q(z)$. A tractable likelihood estimator can be obtained as $R(\omega, z) = p(z, x)p(\omega|z, x)/q(\omega, z)$, where $q(\omega, z)$ is the (tractable) joint density over the "internal randomness" $\omega$ of the Monte Carlo procedure and the final sample $z$, and $p(\omega|z, x)$ is a conditional distribution used to extend the target to also contain these variables. Different choices for the Monte Carlo procedure $q(\omega, z)$ and the target extension $p(\omega|z, x)$ lead to different estimators. To arrive at a particular existing likelihood estimator $R$ requires careful estimator-specific choices and derivations. In contrast, our work proceeds in the opposite direction: we start with an arbitrary estimator $R$ and show (via coupling) how to find a corresponding Monte Carlo procedure $q(\omega, z)$. We also provide a set of tools to "automatically" find couplings for many types of estimators.

The idea of using extended state-spaces is common in (Markov chain) Monte Carlo inference methods [12, 2, 21, 22]. These works also identify extended target distributions that admit $p(z, x)$ as a marginal, i.e., a coupling in our terminology. By running an Markov chain Monte Carlo (MCMC) sampler on the extended target and dropping the auxiliary variables, they obtain an MCMC sampler for $p(z|x)$. Our work can be seen as the VI analogue of these MCMC methods. Other recent work [6, 18, 10, 20, 17, 11, 8, 28] that has explored the connection between using estimators in variational

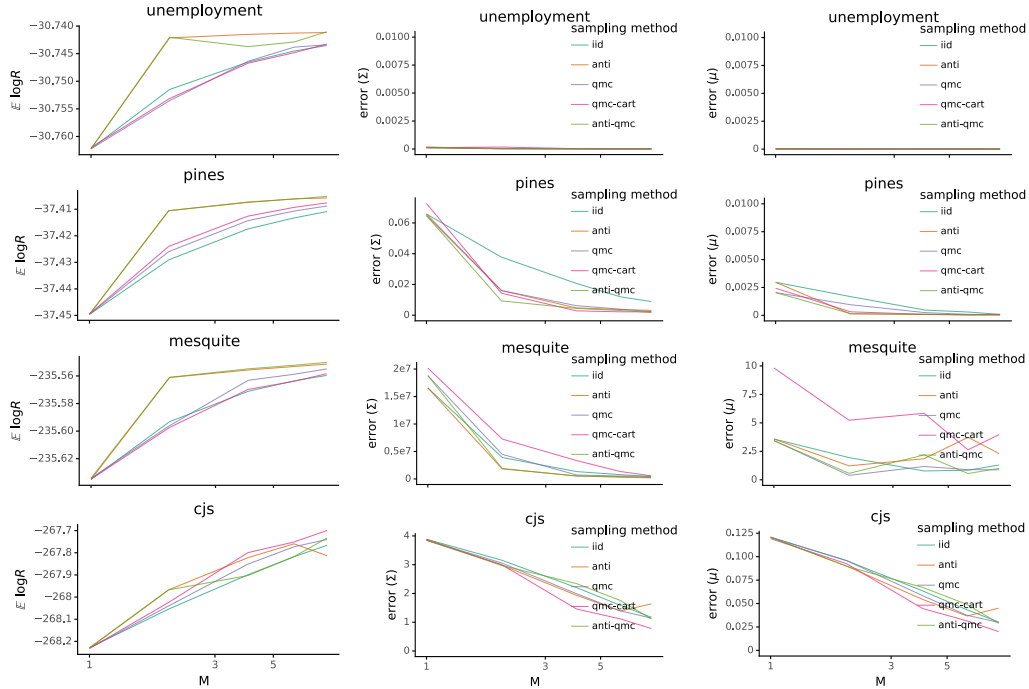

Figure 7: **Across all models, improvements in likelihood bounds correlate strongly with improvements in posterior accuracy**. Better sampling methods can improve both. First row: the common case where a simple Gaussian posterior is already very accurate. Here, only a tiny improvement in the ELBO is possible, and improvement in the posterior is below the level detectable when comparing to MCMC. The other rows show cases where larger improvements are possible.

bounds and auxiliary variational inference [1]. To the best of our knowledge, all of these works consider situations in which the relevant extended state space $(z, \omega)$ is known. Thus, in these works, the estimator essentially comes with an "obvious" coupling distribution $a(z|\omega)$. In contrast, the goal of this paper is to consider an arbitrary estimator $R(\omega)$, where it is not obvious that a tractable coupling distribution $a(z|\omega)$ even exists. This is the situation in which our framework of estimator-coupling pairs is likely to be useful. The alternative would be manual construction of extended state-spaces for each individual estimator.

## Footnotes

[1]With measures instead of densities would write $Q(z \in B, \omega \in A) = \int_A a(z \in B|\omega)dQ(\omega)$ where $a$ is a Markov kernel; see supplement.

[2]With this notation, Eq. (2) becomes $R = \frac{1}{2}\frac{p(\boldsymbol{\omega},x)+p(T(\boldsymbol{\omega}),x)}{Q(\boldsymbol{\omega})}$ where $\boldsymbol{\omega} \sim Q$. Here and in the rest of this paper, when $p(\cdot,\cdot)$ has two arguments, the first always plays the role of $z$.

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
