[Supplementary Material]

# 7 Proofs for Main Framework ([Sec. 3](#))

**Lemma 1.** *Let $\boldsymbol{\omega}$ be a random variable with distribution $Q(\omega)$ and let $R(\boldsymbol{\omega})$ be a positive estimator such that $\mathbb{E}_{Q(\boldsymbol{\omega})} R(\boldsymbol{\omega}) = p(x)$. Then*

$$P^{MC}(\omega, x) \;\; = \;\; Q(\omega)R(\omega)$$

*is an unnormalized distribution over $\omega$ with normalization constant $p(x)$ and $R(\omega) = P^{MC}(\omega, x)/Q(\omega)$ for $Q(\omega) > 0$. Furthermore as defined above,*

$$\log p(x) = \mathbb{E}_{Q(\boldsymbol{\omega})} \log R(\boldsymbol{\omega}) + \mathrm{KL}\left[Q(\boldsymbol{\omega}) \big\| P^{MC}(\boldsymbol{\omega}|x)\right]. \tag{3}$$

*Proof.* Since $P^{\mathrm{MC}}(\omega, x) \geq 0$ and $P^{\mathrm{MC}}(x) = \int P^{\mathrm{MC}}(\omega, x)d\omega = \mathbb{E}_{Q(\boldsymbol{\omega})} R(\boldsymbol{\omega}) = p(x)$, it is a valid distribution. Thus, one can apply the standard ELBO decomposition to $Q(\omega)$ and $P^{\mathrm{MC}}(\omega, x)$. But since $R = P^{\mathrm{MC}}/Q$, it follows that $\mathbb{E}_{Q(\boldsymbol{\omega})} \log \left(P^{\mathrm{MC}}(\boldsymbol{\omega}, x)/Q(\boldsymbol{\omega})\right) = \mathbb{E}_{Q(\boldsymbol{\omega})} \log R(\boldsymbol{\omega})$. $\square$

**Theorem 2.** Suppose that $R(\omega)$ and $a(z|\omega)$ are a valid estimator-coupling pair under $Q(\omega)$. Then,

$$
\begin{aligned}
Q(z, \omega) &\;\;=\;\; Q(\omega)a(z|\omega), & (5) \\
P^{\mathrm{MC}}(z, \omega, x) &\;\;=\;\; Q(\omega)R(\omega)a(z|\omega), & (6)
\end{aligned}
$$

are valid distributions, $P^{\mathrm{MC}}(z, x) = p(z, x)$, and

$$\log p(x) = \mathbb{E}_{Q(\boldsymbol{\omega})} \log R(\boldsymbol{\omega}) + \mathrm{KL}\left[Q(\mathsf{z}) \| p(\mathsf{z}|x)\right] + \mathrm{KL}\left[Q(\boldsymbol{\omega}|\mathsf{z}) \big\| P^{\mathrm{MC}}(\boldsymbol{\omega}|\mathsf{z}, x)\right]. \tag{7}$$

*Proof.* First, note that

$$
\begin{aligned}
P^{\mathrm{MC}}(z, x) &\;\;=\;\; \int P^{\mathrm{MC}}(z, \omega, x)d\omega \\
&\;\;=\;\; \int Q(\omega)R(\omega)a(z|\omega)d\omega \\
&\;\;=\;\; \mathbb{E}_{Q(\boldsymbol{\omega})} R(\boldsymbol{\omega})a(z|\boldsymbol{\omega}) \\
&\;\;=\;\; p(z, x),
\end{aligned}
$$

so $P^{\mathrm{MC}}(z, \omega, x)$ is a valid augmentation of $p(z, x)$.

Next, observe for $P^{\mathrm{MC}}$ and $Q$ as defined,

$$\frac{P^{\mathrm{MC}}(z, \omega, x)}{Q(z, \omega)} = R(\omega).$$

Applying the ELBO decomposition from [Eq. (1)](#) to $Q(z, \omega)$ and $P^{\mathrm{MC}}(z, \omega, x)$ we get that

$$\log P^{\mathrm{MC}}(x) = \mathbb{E}_{Q(\mathsf{z}, \boldsymbol{\omega})} \left[\log \frac{P^{\mathrm{MC}}(\mathsf{z}, \boldsymbol{\omega}, x)}{Q(\mathsf{z}, \boldsymbol{\omega})}\right] + \mathrm{KL}\left[Q(\mathsf{z}, \boldsymbol{\omega}) \big\| P^{\mathrm{MC}}(\mathsf{z}, \boldsymbol{\omega}|x)\right].$$

Using the observations above and the chain rule of KL-divergence means that

$$
\begin{aligned}
\log p(x) &\;\;=\;\; \mathbb{E}_{Q(\boldsymbol{\omega})} \log R(\boldsymbol{\omega}) + \mathrm{KL}\left[Q(\mathsf{z}, \boldsymbol{\omega}) \big\| P^{\mathrm{MC}}(\mathsf{z}, \boldsymbol{\omega}|x)\right] \\
&\;\;=\;\; \mathbb{E}_{Q(\boldsymbol{\omega})} \log R(\boldsymbol{\omega}) + \mathrm{KL}\left[Q(\mathsf{z}) \big\| P^{\mathrm{MC}}(\mathsf{z}|x)\right] + \mathrm{KL}\left[Q(\boldsymbol{\omega}|\mathsf{z}) \big\| P^{\mathrm{MC}}(\boldsymbol{\omega}|\mathsf{z}, x)\right] \\
&\;\;=\;\; \mathbb{E}_{Q(\boldsymbol{\omega})} \log R(\boldsymbol{\omega}) + \mathrm{KL}\left[Q(\mathsf{z}) \| p(\mathsf{z}|x)\right] + \mathrm{KL}\left[Q(\boldsymbol{\omega}|\mathsf{z}) \big\| P^{\mathrm{MC}}(\boldsymbol{\omega}|\mathsf{z}, x)\right].
\end{aligned}
$$

$\square$

**Claim 5.** *Suppose that $Q(T(\omega)) = Q(\omega)$. Then, the antithetic estimator*

$$R(\omega) = \frac{p(\omega, x) + p(T(\omega), x)}{2Q(\omega)}$$

*and the coupling distribution*

$$
\begin{aligned}
a(z|\omega) &= \pi(\omega)\,\delta(z - \omega) + (1 - \pi(\omega))\,\delta(z - T(\omega)), \\
\pi(\omega) &= \frac{p(\omega, x)}{p(\omega, x) + p(T(\omega), x)}.
\end{aligned}
$$

*form a valid estimator / coupling pair under $Q(\omega)$.*

*Proof.*

$$
\begin{aligned}
&\underset{Q(\boldsymbol{\omega})}{\mathbb{E}}\, R(\boldsymbol{\omega})a(z|\boldsymbol{\omega}) \\
&= \underset{Q(\boldsymbol{\omega})}{\mathbb{E}}\, \frac{p(\boldsymbol{\omega}, x) + p(T(\boldsymbol{\omega}), x)}{2Q(\boldsymbol{\omega})}\left(\pi(\boldsymbol{\omega})\,\delta(z - \boldsymbol{\omega}) + (1 - \pi(\boldsymbol{\omega}))\,\delta(z - T(\boldsymbol{\omega}))\right) \\
&= \underset{Q(\boldsymbol{\omega})}{\mathbb{E}}\, \frac{p(\boldsymbol{\omega}, x) + p(T(\boldsymbol{\omega}), x)}{2Q(\boldsymbol{\omega})}\Big(\frac{p(\boldsymbol{\omega}, x)}{p(\boldsymbol{\omega}, x) + p(T(\boldsymbol{\omega}), x)}\,\delta(z - \boldsymbol{\omega}) \\
&\qquad\qquad\qquad + \frac{p(T(\boldsymbol{\omega}), x)}{p(\boldsymbol{\omega}, x) + p(T(\boldsymbol{\omega}), x)}\,\delta(z - T(\boldsymbol{\omega}))\Big) \\
&= \underset{Q(\boldsymbol{\omega})}{\mathbb{E}}\, \frac{1}{2Q(\boldsymbol{\omega})}\left(p(\boldsymbol{\omega}, x)\,\delta(z - \boldsymbol{\omega}) + p(T(\boldsymbol{\omega}), x)\,\delta(z - T(\boldsymbol{\omega}))\right) \\
&= \underset{Q(\boldsymbol{\omega})}{\mathbb{E}}\, \frac{1}{2}\left(\frac{1}{Q(\boldsymbol{\omega})}p(\boldsymbol{\omega}, x)\,\delta(z - \boldsymbol{\omega}) + \frac{1}{Q(\boldsymbol{\omega})}p(T(\boldsymbol{\omega}), x)\,\delta(z - T(\boldsymbol{\omega}))\right) \\
&= \underset{Q(\boldsymbol{\omega})}{\mathbb{E}}\, \frac{1}{2}\left(\frac{1}{Q(\boldsymbol{\omega})}p(\boldsymbol{\omega}, x)\,\delta(z - \boldsymbol{\omega}) + \frac{1}{Q(T(\boldsymbol{\omega}))}p(T(\boldsymbol{\omega}), x)\,\delta(z - T(\boldsymbol{\omega}))\right) \qquad (10) \\
&= \underset{Q(\boldsymbol{\omega})}{\mathbb{E}}\, \frac{1}{2}\left(\frac{1}{Q(\boldsymbol{\omega})}p(\boldsymbol{\omega}, x)\,\delta(z - \boldsymbol{\omega}) + \frac{1}{Q(\boldsymbol{\omega})}p(\boldsymbol{\omega}, x)\,\delta(z - \boldsymbol{\omega})\right) \qquad (11) \\
&= \underset{Q(\boldsymbol{\omega})}{\mathbb{E}}\, \left(\frac{1}{Q(\boldsymbol{\omega})}p(\boldsymbol{\omega}, x)\,\delta(z - \boldsymbol{\omega})\right) \\
&= \int \left(p(\omega, x)\,\delta(z - \omega)\right) d\omega \\
&= p(z, x)
\end{aligned}
$$

Here, Eq. (10) follows from the fact that $Q(T(\omega)) = Q(\omega)$ while Eq. (11) follows from the fact that $T(\boldsymbol{\omega})$ is equal in distribution to $\boldsymbol{\omega}$ when $\boldsymbol{\omega} \sim Q$. $\qquad\square$

# 8 Measure-Theoretic Details

The content of this section draws from [14, 15]. We do not use sans-serif font in this section.

## 8.1 Measures, KL, ELBO

Let $(\Omega, \mathcal{A})$ be a measurable space and $Q$ and $P$ be two measures over it. Write $Q \ll P$ when $Q$ is absolutely continuous with respect to $P$, i.e. when $P(A) = 0 \Rightarrow Q(A) = 0$. Whenever $Q \ll P$, there exists measurable $f : \Omega \to \mathbb{R}$ such that

$$Q(A) = \int_A f \, dP.$$

The function $f$ is the *Radon-Nikodym derivative,* denoted as $f = \frac{dQ}{dP}$. Write $Q \sim P$ when $Q \ll P$ and $P \ll Q$; in this case $\frac{dQ}{dP} = \left( \frac{dP}{dQ} \right)^{-1}$ $Q$-a.e.

For two probability measures $Q \ll P$, the KL-divergence is

$$\mathrm{KL}\left[Q\|P\right] = \int \log \left( \frac{dQ}{dP} \right) Q(d\omega) = \underset{Q(\omega)}{\mathbb{E}} \log \frac{dQ}{dP}.$$

For a probability measure $Q$ and measure $\hat{P}$ (not necessarily a probability measure) with $Q \ll \hat{P}$, the evidence lower bound or "ELBO" is

$$\mathrm{ELBO}\left[Q\middle\|\hat{P}\right] = - \underset{Q}{\mathbb{E}} \log \frac{dQ}{d\hat{P}}.$$

When $Q \sim \hat{P}$, we can equivalently write $\mathrm{ELBO}\left[Q\middle\|\hat{P}\right] = \mathbb{E}_Q \log \frac{d\hat{P}}{dQ}$.

Let $(Z, \mathcal{B})$ be a measurable space. Let $P_{z,x}$ be an unnormalized distribution over $z$ representing the joint distribution over $(z, x)$ for a fixed $x$. Write either $P_{z,x}(B)$ or $P_{z,x}(z \in B)$ for the measure of $B \in \mathcal{B}$. Define

$$p(x) = P_{z,x}(Z)$$

to be the total measure or the normalization constant of $P_{z,x}$, and write $P_{z|x}(z \in B) := P_{z,x}(z \in B)/p(x)$ for the corresponding normalized measure, which represents the conditional distribution of $z$ given $x$. Henceforth, $x$ will *always* denote a fixed constant, and, for any $u$, the measure $P_{u,x}$ is unnormalized with total measure $p(x)$.

The following gives a measure-theoretic version of the "ELBO decomposition" from Eq. (1).

**Lemma 6.** *Given a probability measure $Q$ and a measure $P_{z,x}$ on $(Z, \mathcal{B})$, whenever $Q \ll P_{z,x}$ we have the following "ELBO decomposition":*

$$\log p(x) = \mathrm{ELBO}\left[Q\|P_{z,x}\right] + \mathrm{KL}\left[Q\middle\|P_{z|x}\right].$$

*Proof.* It is easy to check that $\frac{dQ}{dP_{z|x}} = p(x) \frac{dQ}{dP_{z,x}}$.[3] Then

$$\mathrm{KL}\left[Q\middle\|P_{z|x}\right] = \underset{Q}{\mathbb{E}} \log \frac{dQ}{dP_{z|x}} = \underset{Q}{\mathbb{E}} \log \left( p(x) \frac{dQ}{dP_{z,x}} \right)$$

$$= \log p(x) + \underset{Q}{\mathbb{E}} \log \frac{dQ}{dP_{z,x}} = \log p(x) - \mathrm{ELBO}\left[Q\|P_{z,x}\right].$$

Rearranging, we see the ELBO decomposition. □

## 8.2 Conditional, Marginal, and Joint Distributions

**Standard Borel and product spaces**    We will assume that each relevant measure space is a *standard Borel space*, that is, isomorphic to a Polish space (a separable complete metric space) with the Borel $\sigma$-algebra. Standard Borel spaces capture essentially all spaces that arise in practice in probability theory [15]. Let $(\Omega, \mathcal{A})$ and $(Z, \mathcal{B})$ be standard Borel spaces. The *product space* $(\Omega \times Z, \mathcal{A} \otimes \mathcal{B})$ is the measurable space on $\Omega \times Z$ with $\mathcal{A} \otimes \mathcal{B} = \{A \times B : A \in \mathcal{A}, B \in \mathcal{B}\}$, and is also a standard Borel space.

**Conditional distributions**    We require tools to augment a distribution with a new random variable and define the conditional distribution of one random variable with respect to another. We begin with a Markov kernel, which we will use to augment a distribution $P_\omega$ with a new random variable to obtain a joint distribution $P_{\omega,z}$.

Formally, a *Markov kernel [15, Def. 8.24]* from $(\Omega, \mathcal{A})$ to $(Z, \mathcal{B})$ is a mapping $a(B|\omega)$ that satisfies:

1. For fixed $\omega$, $a(B|\omega)$ is a probability measure on $(Z, \mathcal{B})$.

2. For fixed $B$, $a(B|\omega)$ is an $\mathcal{A}$-measurable function of $\omega$.

Let $P_\omega$ be a measure on $(\Omega, A)$ and $a(B|\omega)$ a Markov kernel from $(\Omega, A)$ to $(Z, \mathcal{B})$. These define a unique measure $P_{\omega,z}$ over the product space defined as

$$P_{\omega,z}(\omega \in A, z \in B) = \int_A a(z \in B|\omega) P_\omega(d\omega),$$

such that if $P_\omega$ is a probability measure, then $P_{\omega,z}$ is also a probability measure [15, Cor. 14.23].

Alternately, we may have a joint distribution $P_{\omega,z}$ (a measure on the product space $(\Omega \times Z, \mathcal{A} \otimes \mathcal{B})$) and want to define the marginals and conditionals. The *marginal distribution $P_z$* is the measure on $(Z, \mathcal{B})$ with $P_z(z \in B) = P_{\omega,z}(\omega \in \Omega, z \in B)$, and the marginal $P_\omega$ is defined analogously. Since the product space is standard Borel [15, Thm 14.8], there exists a *regular conditional distribution $P_{\omega|z}(\omega \in A|z)$* [15, Def. 8.27, Thm. 8.36], which is a Markov kernel (as above) and satisfies the following for all $A \in \mathcal{A}$, $B \in \mathcal{B}$:

$$P_{\omega,z}(\omega \in A, z \in B) = \int_B P_{\omega|z}(\omega \in A|z) P_z(dz).$$

The regular conditional distribution is unique up to null sets of $P_z$.

The conditional distribution $P_{z|\omega}$ is defined analogously.

## 8.3 KL Chain Rule

Let $P_{\omega,z}$ and $Q_{\omega,z}$ be two probability measures on the standard Borel product space $(\Omega \times Z, \mathcal{A} \otimes \mathcal{B})$ with $Q_{\omega,z} \ll P_{\omega,z}$. The *conditional KL-divergence* $\text{KL}\left[Q_{\omega|z}\middle\|P_{\omega|z}\right]$ is defined[4] as [14, Ch. 5.3]

$$\text{KL}\left[Q_{\omega|z}\middle\|P_{\omega|z}\right] = \mathop{\mathbb{E}}_{Q_{\omega,z}}\left(\frac{dQ_{\omega|z}}{dP_{\omega|z}}\right),$$

where $\frac{dQ_{\omega|z}}{dP_{\omega|z}}(\omega|z) = \left(\frac{dQ_{\omega,z}}{dP_{\omega,z}}(\omega, z)\right)\left(\frac{dQ_z}{dP_z}(z)\right)^{-1}$ when $\frac{dQ_z}{dP_z}(z) > 0$ and 1 otherwise. When all densities exist, $\frac{dQ_{\omega|z}}{dP_{\omega|z}}(\omega|z) = \frac{q(\omega|z)}{p(\omega|z)}$. Under the same conditions as above, we have the *chain rule for KL-divergence [14, Lem. 5.3.1]*

$$\text{KL}\left[Q_{\omega,z}\|P_{\omega,z}\right] = \text{KL}\left[Q_\omega\|P_\omega\right] + \text{KL}\left[Q_{\omega|z}\middle\|P_{\omega|z}\right] = \text{KL}\left[Q_z\|P_z\right] + \text{KL}\left[Q_{z|\omega}\middle\|P_{z|\omega}\right].$$

## 8.4 Our Results

Now consider a strictly positive estimator $R(\omega)$ over probability space $(\Omega, \mathcal{A}, Q_\omega)$ such that $\mathbb{E}_{Q_\omega} R = \int R \, dQ_\omega = p(x)$. We wish to define $P_{\omega,x}^{\mathrm{MC}}$ so that $\frac{dP_{\omega,x}^{\mathrm{MC}}}{dQ_\omega} = R$, to justify interpreting $\mathbb{E}_{Q_\omega} \log R$ as an ELBO. This is true when $R = \frac{dP_{\omega,x}^{\mathrm{MC}}}{dQ_\omega}$ is the Radon-Nikodym derivative, i.e., a change of measure from $Q_\omega$ to $P_{\omega,x}^{\mathrm{MC}}$, and is strictly positive. This leads to the definition

$$P_{\omega,x}^{\mathrm{MC}}(\omega \in A) = \int_A R \, dQ_\omega.$$

**Lemma 7.** *Let $R(\omega)$ be an almost-everywhere positive random variable on $(\Omega, \mathcal{A}, Q_\omega)$ with $\mathbb{E}_{Q_\omega} R = p(x)$ and define $P_{\omega,x}^{\mathrm{MC}}(\omega \in A) = \int_A R \, dQ_\omega$. The ELBO decomposition applied to $Q_\omega$ and $P_{\omega,x}^{\mathrm{MC}}$ gives:*

$$\log p(x) = \mathbb{E}_{Q_\omega} \log R + \mathrm{KL}\left[Q_\omega \middle\| P_{\omega|x}^{\mathrm{MC}}\right].$$

*Proof.* By construction, $R = \frac{dP_{\omega,x}^{\mathrm{MC}}}{dQ_\omega}$ and $P_{\omega,x}^{\mathrm{MC}} \sim Q_\omega$, since $R$ is positive $Q$-a.e. Therefore $\mathbb{E}_{Q_\omega} \log R = \mathbb{E}_{Q_\omega} \log \frac{dP_{\omega,x}^{\mathrm{MC}}}{dQ} = \mathrm{ELBO}\left[Q \middle\| P_{\omega,x}^{\mathrm{MC}}\right]$, where the final equality uses the definition of the ELBO for the case when $P_{\omega,x}^{\mathrm{MC}} \sim Q_\omega$. Now apply Lem. 6 and the fact that $\mathrm{ELBO}\left[Q \middle\| P_{\omega,x}^{\mathrm{MC}}\right] = \mathbb{E}_{Q_\omega} \log R$. $\qquad\square$

Lem. 7 provides distributions $Q_\omega$ and $P_{\omega,x}^{\mathrm{MC}}$ so that $\mathbb{E}_{Q_\omega} \log R = \mathrm{ELBO}\left[Q_\omega \middle\| P_{\omega,x}^{\mathrm{MC}}\right]$, which justifies maximizing the likelihood bound $\mathbb{E}_{Q_\omega} \log R$ as minimzing the KL-divergence from $Q_\omega$ to the "target" $P_{\omega|x}^{\mathrm{MC}}$. However, neither distribution contains the random variable $z$ from the original target distribution $P_{z|x}$, so the significance of Lem. 7 on its own is unclear. We now describe a way to couple $P_{\omega,x}^{\mathrm{MC}}$ to the original target distribution using a Markov kernel $a(z \in B|\omega)$.

**Definition 8.** A valid *estimator-coupling pair* with respect to target distribution $P_{z,x}$ is an estimator $R(\omega)$ on probability space $(\Omega, \mathcal{A}, Q_\omega)$ and Markov kernel $a(z \in B|\omega)$ from $(\Omega, \mathcal{A})$ to $(Z, \mathcal{B})$ such that:

$$\mathbb{E}_{Q_\omega} R(\omega) a(z \in B|\omega) = P_{z,x}(z \in B).$$

**Lemma 9.** *Assume $R(\omega)$ and $a(z \in B|\omega)$ are a valid estimator-coupling pair with respect to target $P_{z,x}$, and define*

$$P_{\omega,z,x}^{\mathrm{MC}}(\omega \in A, z \in B) = \int_A a(z \in B|\omega) \, R(\omega) \, Q_\omega(d\omega).$$

*Then $P_{\omega,z,x}^{\mathrm{MC}}$ admits $P_{z,x}$ as a marginal, i.e., $P_{z,x}^{\mathrm{MC}}(z \in B) = P_{z,x}(z \in B)$.*

*Proof.* We have

$$\begin{aligned}
P_{z,x}^{\mathrm{MC}}(z \in B) &= P_{\omega,z,x}^{\mathrm{MC}}(\omega \in \Omega, z \in B) \\
&= \int_\Omega a(z \in B|\omega) R(\omega) Q_\omega(d\omega). \\
&= \mathbb{E}_{Q_\omega} R(\omega) a(z \in B|\omega) \\
&= P_{z,x}(z \in B).
\end{aligned}$$

The second line uses the definition of $P_{\omega,z,x}^{\mathrm{MC}}$. The last line uses the definition of a valid estimator-coupling pair. $\qquad\square$

**Theorem 10.** *Let $P_{z,x}$ be an unnormalized distribution with normalization constant $p(x)$. Assume $R(\omega)$ and $a(z \in B|\omega)$ are a valid estimator-coupling pair with respect to $P_{z,x}$. Define $P_{\omega,z,x}^{\mathrm{MC}}$ as in Lem. 9 and define $Q_{\omega,z}(\omega \in A, z \in B) = \int_A a(z \in B|\omega) \, Q_\omega(d\omega)$. Then*

$$\log p(x) = \mathop{\mathbb{E}}_{Q_\omega} \log R + \mathrm{KL}\left[Q_z \middle\| P_{z|x}\right] + \mathrm{KL}\left[Q_{\omega|z} \middle\| P_{\omega|z,x}^{\mathrm{MC}}\right].$$

*Proof.* From Lem. 7, we have

$$\log p(x) = \mathop{\mathbb{E}}_{Q_\omega} \log R + \mathrm{KL}\left[Q_\omega \middle\| P_{\omega|x}^{\mathrm{MC}}\right].$$

We will show by two applications of the KL chain rule that the second term can be expanded as

$$\mathrm{KL}\left[Q_\omega \middle\| P_{\omega|x}^{\mathrm{MC}}\right] = \mathrm{KL}\left[Q_z \middle\| P_{z|x}\right] + \mathrm{KL}\left[Q_{\omega|z} \middle\| P_{\omega|z,x}^{\mathrm{MC}}\right], \tag{12}$$

which will complete the proof.

We first apply the KL chain rule as follows:

$$\mathrm{KL}\left[Q_{\omega,z} \middle\| P_{\omega,z|x}^{\mathrm{MC}}\right] = \mathrm{KL}\left[Q_\omega \middle\| P_{\omega|x}^{\mathrm{MC}}\right] + \underbrace{\mathrm{KL}\left[Q_{z|\omega} \middle\| P_{z|\omega,x}^{\mathrm{MC}}\right]}_{=0}. \tag{13}$$

We now argue that the second term is zero, as indicated in the equation. Note from above that $\frac{dP_{\omega,x}^{\mathrm{MC}}}{dQ_\omega} = R$. It is also true that $\frac{dP_{\omega,z,x}^{\mathrm{MC}}}{dQ_{\omega,z}} = R$. To see this, observe that

$$
\begin{aligned}
\int_{A \times B} R(\omega)\, Q_{\omega,z}(d\omega, dz) &= \int_A \left( \int_B R(\omega) a(z \in dz|\omega) \right) Q_\omega(d\omega) \\
&= \int_A R(\omega) \left( \int_B a(z \in dz|\omega) \right) Q_\omega(d\omega) \\
&= \int_A R(\omega)\, a(z \in B|\omega)\, Q_\omega(d\omega) \\
&= P_{\omega,z,x}^{\mathrm{MC}}(\omega \in A, z \in B).
\end{aligned}
$$

The first equality above uses a version of Fubini's theorem for Markov kernels [**?** , Thm. 14.29]. Because $P_{\omega,x}^{\mathrm{MC}} \sim Q_\omega$ it also follows that $\frac{dQ_\omega}{dP_{\omega,x}^{\mathrm{MC}}} = \frac{dQ_{\omega,z}}{dP_{\omega,z,x}^{\mathrm{MC}}} = \frac{1}{R}$. Since the normalized distributions $P_{\omega|x}^{\mathrm{MC}}$ and $P_{\omega,z|x}^{\mathrm{MC}}$ differ from the unnormalized counterparts by the constant factor $p(x)$, it is straightforward to see that $\frac{dQ_\omega}{dP_{\omega|x}^{\mathrm{MC}}} = \frac{dQ_{\omega,z}}{dP_{\omega,z|x}^{\mathrm{MC}}} = \frac{p(x)}{R}$.[5] This implies that $\frac{dP_{z|\omega,x}^{\mathrm{MC}}}{dQ_{z|\omega}} = 1$ a.e., which in turn implies that the conditional divergence $\mathrm{KL}\left[Q_{z|\omega} \middle\| P_{z|\omega,x}^{\mathrm{MC}}\right]$ is equal to zero.

We next apply the chain rule the other way and use the fact that $P_{z|x}^{\mathrm{MC}} = P_{z|x}$ (Lem. 9) to see that:

$$\mathrm{KL}\left[Q_{\omega,z} \middle\| P_{\omega,z|x}^{\mathrm{MC}}\right] = \mathrm{KL}\left[Q_z \middle\| P_{z|x}^{\mathrm{MC}}\right] + \mathrm{KL}\left[Q_{\omega|z} \middle\| P_{\omega|z,x}^{\mathrm{MC}}\right] = \mathrm{KL}\left[Q_z \middle\| P_{z|x}\right] + \mathrm{KL}\left[Q_{\omega|z} \middle\| P_{\omega|z,x}^{\mathrm{MC}}\right]. \tag{14}$$

Combining Eq. (13) and Eq. (14) we get Eq. (12), as desired.

$\square$

# 9 Specific Variance Reduction Techniques

## 9.1 IID Mean

As a simple example, consider the IID mean. Suppose $R_0(\omega)$ and $a_0(z|\omega)$ are valid under $Q_0$. If we define

$$Q(\omega_1, \cdots, \omega_M, m) = \frac{1}{M} \prod_{m=1}^{M} Q_0(\omega_m)$$

(with $\boldsymbol{\omega}_1, \cdots, \boldsymbol{\omega}_M \sim Q_0$ i.i.d. and $\mathsf{m}$ uniform on $\{1, \cdots, M\}$) then this satisfies the condition of Thm. 3 that $\boldsymbol{\omega}_{\mathsf{m}} \sim Q_0$. Thus we can define $R$ and $a$ as in Eq. (8) and Eq. (9), to get that

$$
\begin{aligned}
R(\omega_1, \cdots, \omega_M, m) &= R_0(\omega_m) \\
a(z|\omega_1, \cdots, \omega_M, m) &= a_0(z|\omega_m)
\end{aligned}
$$

are a valid estimator-coupling pair under $Q$. Note that $Q(m|\omega_1, \cdots, \omega_M) = \frac{1}{M}$, so if we apply Thm. 4 to marginalize out $m$, we get that

$$
\begin{aligned}
R(\omega_1, \cdots, \omega_M) &= \underset{Q(\mathsf{m}|\boldsymbol{\omega}_1, \cdots, \boldsymbol{\omega}_M)}{\mathbb{E}} R(\omega_1, \cdots, \omega_M, \mathsf{m}) \\
&= \frac{1}{M} \sum_{m=1}^{M} R(\omega_1, \cdots, \omega_M, m) \\
&= \frac{1}{M} \sum_{m=1}^{M} R_0(\omega_m) \\
a(z|\omega_1, \cdots, \omega_M) &= \frac{1}{R(\omega_1, \cdots, \omega_M)} \underset{Q(\mathsf{m}|\boldsymbol{\omega}_1, \cdots, \omega_M)}{\mathbb{E}} [R(\omega_1, \cdots, \omega_M, \mathsf{m}) \, a(z|\omega_1, \cdots, \omega_M, \mathsf{m})] \\
&= \frac{1}{R(\omega_1, \cdots, \omega_M)} \frac{1}{M} \sum_{m=1}^{M} [R(\omega_1, \cdots, \omega_M, m) \, a(z|\omega_1, \cdots, \omega_M, m)] \\
&= \frac{1}{\frac{1}{M} \sum_{m=1}^{M} R_0(\omega_m)} \frac{1}{M} \sum_{m=1}^{M} [R_0(\omega_m) a_0(z|\omega_m)] \\
&= \frac{\sum_{m=1}^{M} [R_0(\omega_m) a_0(z|\omega_m)]}{\sum_{m=1}^{M} R_0(\omega_m)}.
\end{aligned}
$$

These are exactly the forms for $R(\cdot)$ and $a(z|\cdot)$ shown in the table.

## 9.2 Stratified Sampling

As another example, take stratified sampling. The estimator-coupling pair can be derived similiarly to with the i.i.d. mean. For simplicity, we assume here one sample in each strata ($N_m = 1$). Suppose $\Omega_1 \cdots \Omega_M$ partition the state-space and define

$$Q(\omega_1, \cdots, \omega_M, m) = \frac{1}{M} \prod_{k=1}^{M} \frac{Q_0(\omega_k) I(\omega_k \in \Omega_m)}{\mu(k)} \times \mu(m), \quad \mu(m) = \underset{Q_0(\boldsymbol{\omega})}{\mathbb{E}} I(\boldsymbol{\omega} \in \Omega_m).$$

This again satisfies the condition of Thm. 3, so Eq. (8) and Eq. (9) give that

$$
\begin{aligned}
R(\omega_1, \cdots, \omega_M, m) &= R_0(\omega_m) \\
a(z|\omega_1, \cdots, \omega_M, m) &= a_0(z|\omega_m)
\end{aligned}
$$

is a valid estimator-coupling pair with respect to $Q$. Note that $Q(\mathsf{m}|\omega_1, \cdots, \omega_M) = \mu(m)$, so if we apply Thm. 4 to marginalize out $m$, we get that

$$
\begin{aligned}
R(\omega_1, \cdots, \omega_M) &= \underset{Q(\mathsf{m}|\omega_1,\cdots,\omega_M)}{\mathbb{E}} R(\omega_1, \cdots, \omega_M, \mathsf{m}) \\
&= \sum_{m=1}^{M} \mu(m)\, R(\omega_1, \cdots, \omega_M, m) \\
&= \sum_{m=1}^{M} \mu(m) R_0(\omega_m) \\
a(z|\omega_1, \cdots, \omega_M) &= \frac{1}{R(\omega_1, \cdots, \omega_M)} \underset{Q(\mathsf{m}|\omega_1,\cdots,\omega_M)}{\mathbb{E}} \left[ R(\omega_1, \cdots, \omega_M, \mathsf{m})\, a(z|\omega_1, \cdots, \omega_M, \mathsf{m}) \right] \\
&= \frac{1}{R(\omega_1, \cdots, \omega_M)} \sum_{m=1}^{M} \mu(m) R(\omega_1, \cdots, \omega_M, m)\, a(z|\omega_1, \cdots, \omega_M, m) \\
&= \frac{\sum_{m=1}^{M} \mu(m) R_0(\omega_m) a_0(z|\omega_m)}{\sum_{m=1}^{M} \mu(m) R_0(\omega_m)}.
\end{aligned}
$$

Again, this is the form shown in the table.

# 10 Proofs for Deriving Couplings (Sec. 4)

**Theorem 3.** Suppose that $R_0(\omega)$ and $a_0(z|\omega)$ are a valid estimator-coupling pair under $Q_0(\omega)$. Let $Q(\omega_1, \cdots, w_M, m)$ be any distribution such that if $(\boldsymbol{\omega}_1, \cdots, \boldsymbol{\omega}_M, \mathsf{m}) \sim Q$, then $\boldsymbol{\omega}_\mathsf{m} \sim Q_0$. Then,

$$R(\omega_1, \cdots, \omega_M, m) = R_0(\omega_m) \tag{8}$$

$$a(z|\omega_1, \cdots, \omega_M, m) = a_0(z|\omega_m) \tag{9}$$

are a valid estimator-coupling pair under $Q(\omega_1, \cdots, w_M, m)$.

*Proof.* Substitute the definitions of $R$ and $a$ to get that

$$
\begin{aligned}
\mathop{\mathbb{E}}_{Q(\boldsymbol{\omega}_1, \cdots, \boldsymbol{\omega}_M, \mathsf{m})} R(\omega_1, \cdots, \omega_M, m) a(z|\omega_1, \cdots, \omega_M, m) &= \mathop{\mathbb{E}}_{Q(\boldsymbol{\omega}_1, \cdots, \boldsymbol{\omega}_M, \mathsf{m})} R_0(\boldsymbol{\omega}_\mathsf{m}) a_0(z|\boldsymbol{\omega}_\mathsf{m}) \\
&= \mathop{\mathbb{E}}_{Q_0(\boldsymbol{\omega})} R_0(\boldsymbol{\omega}) a_0(z|\boldsymbol{\omega}) \\
&= p(z, x),
\end{aligned}
$$

which is equivalent to the definition of $R$ and $a$ being a valid estimator-coupling pair. The second line follows from the assumption on $Q(\omega_1, \cdots, \omega_M, m)$. $\square$

**Theorem 4.** Suppose that $R_0(\omega, \nu)$ and $a_0(z|\omega, \nu)$ are a valid estimator-coupling pair under $Q_0(\omega, \nu)$. Then

$$R(\omega) = \mathop{\mathbb{E}}_{Q_0(\boldsymbol{\nu}|\omega)} R_0(\omega, \boldsymbol{\nu}),$$

$$a(z|\omega) = \frac{1}{R(\omega)} \mathop{\mathbb{E}}_{Q_0(\boldsymbol{\nu}|\omega)} [R_0(\omega, \boldsymbol{\nu}) a_0(z|\omega, \boldsymbol{\nu})],$$

are a valid estimator-coupling pair under $Q(\omega) = \int Q_0(\omega, \nu) d\nu$.

*Proof.* Substitute the definition of $a$ to get that

$$
\begin{aligned}
\mathop{\mathbb{E}}_{Q(\boldsymbol{\omega})} R(\boldsymbol{\omega}) a(z|\boldsymbol{\omega}) &= \mathop{\mathbb{E}}_{Q(\boldsymbol{\omega})} R(\boldsymbol{\omega}) \frac{1}{R(\boldsymbol{\omega})} \mathop{\mathbb{E}}_{Q_0(\boldsymbol{\nu}|\boldsymbol{\omega})} [R_0(\boldsymbol{\omega}, \boldsymbol{\nu}) a_0(z|\boldsymbol{\omega}, \boldsymbol{\nu})] \\
&= \mathop{\mathbb{E}}_{Q_0(\boldsymbol{\omega}, \boldsymbol{\nu})} [R_0(\boldsymbol{\omega}, \boldsymbol{\nu}) a_0(z|\boldsymbol{\omega}, \boldsymbol{\nu})] \\
&= p(z, x),
\end{aligned}
$$

which is equivalent to $R$ and $a$ being a valid estimator-coupling pair under $R(\omega)$. The last line follows from the fact that $R_0$ and $a_0$ are a valid estimator-coupling pair under $Q_0$. $\square$

Figure 8: The target density $p(z|x)$ and the approximation $Q(z|x)$ produced by various sampling methods (row) with various $M$ (columns). The dark curves show isocontours of kernel density estimate for samples generated using Stan and projected to the first two principal components. The darker curves show isocontours for the process applied to samples from $Q(z|x)$. Antithetic sampling is visibly (but subtly) better than iid for $M = 2$ while the combination of quasi-Monte Carlo and antithetic sampling is (still more subtly) best for $M = 8$.

## 11 Additional Experimental Results

Fig. 9 shows additional aggregate statistics of ELBO and posterior variance error for different methods across model from the Stan library.

Figure 9: **How much do methods improve over naive VI in likelihood bound (x-axis) and in estimating posterior variance (y-axis)?** Each point corresponds to a model from the Stan library, with a random shape. Each plot compares `iid` sampling against some other strategy. From top, these are antithetic sampling (`anti`), Quasi-Monte Carlo, either using an elliptical mapping (`qmc`) or a Cartesian mapping (`qmc-cart`), and antithetic sampling after an elliptical mapping (`anti-qmc`). The columns correspond to using $M = 2, 4$ and $8$ samples for each estimate. **Conclusions**: Improvements in ELBO and error are correlated. Improvements converge to those of `iid` for larger $M$, as all errors decay towards zero. Different sampling methods are best on different datasets. A few cases are not plotted where the measured "improvement" was negative (if naive VI has near-zero error, or due to local optima).

## 12 Full Results For All Models

Figure 10: **Across all models, improvements in likelihood bounds correlate strongly with improvements in posterior accuracy. Better sampling methods can improve both.**

Figure 11: **Across all models, improvements in likelihood bounds correlate strongly with improvements in posterior accuracy. Better sampling methods can improve both.**

Figure 12: **Across all models, improvements in likelihood bounds correlate strongly with improvements in posterior accuracy. Better sampling methods can improve both.**

Figure 13: **Across all models, improvements in likelihood bounds correlate strongly with improvements in posterior accuracy. Better sampling methods can improve both.**

Figure 14: **Across all models, improvements in likelihood bounds correlate strongly with improvements in posterior accuracy. Better sampling methods can improve both.**

Figure 15: **Across all models, improvements in likelihood bounds correlate strongly with improvements in posterior accuracy. Better sampling methods can improve both.**

Figure 16: **Across all models, improvements in likelihood bounds correlate strongly with improvements in posterior accuracy. Better sampling methods can improve both.**

Figure 17: **Across all models, improvements in likelihood bounds correlate strongly with improvements in posterior accuracy. Better sampling methods can improve both.**

Figure 18: **Across all models, improvements in likelihood bounds correlate strongly with improvements in posterior accuracy. Better sampling methods can improve both.**

Figure 19: **Across all models, improvements in likelihood bounds correlate strongly with improvements in posterior accuracy. Better sampling methods can improve both.**

Figure 20: **Across all models, improvements in likelihood bounds correlate strongly with improvements in posterior accuracy. Better sampling methods can improve both.**

Figure 21: **Across all models, improvements in likelihood bounds correlate strongly with improvements in posterior accuracy. Better sampling methods can improve both.**

Figure 22: **Across all models, improvements in likelihood bounds correlate strongly with improvements in posterior accuracy. Better sampling methods can improve both.**

## Footnotes

[3] $\int_A p(x) \frac{dQ}{dP_{z,x}} dP_{z|x} = \int_A \frac{dQ}{dP_{z,x}} dP_{z,x} = Q(z \in A)$.

[4] While this (standard) notation for the divergence refers to "$Q_{\omega|z}$" it is a function of the joint $Q_{\omega,z}$ and similiarly for $P_{\omega,z}$.

[5] $\int_A \frac{p(x)}{R} dP_{\omega|x}^{\mathrm{MC}} = \int_A \frac{1}{R} dP_{\omega,x}^{\mathrm{MC}} = Q_\omega(\omega \in A)$, and similarly for $Q_{\omega,z}$ and $P_{\omega,z,x}^{\mathrm{MC}}$.