[Reviews · NeurIPS 2019]

Reviewer 1



This paper contributes methodology for the improvement of methods that are commonly used for statistical, likelihood-based, approximate inference. More precisely, the paper propose improvements of the popular Variational Inference method. The authors investigate the optimisation of variational objectives constructed by using Monte Carlo estimators of the likelihood and particularly the effect of such optimisations on the approximation of the posterior distribution of interest. They develop a procedure that they call “divide and couple” in order to find an approximation to the posterior of interest that diverges no more than the gap between the log-likelihood and its lower bound derived from the Monte Carlo estimator. The authors give both the theoretical foundations for their new method and guidelines for its practical implementation. I find the research question very interesting and the paper clearly written with careful description of their new methodology. I have below a few comments on the paper. 1) The authors have to organise better Sections 4 and 5 in order the implementation and the results of their empirical study to be more clearly presented. Moreover, a small section with conclusions would very useful if added after Section 5. 2) There are small mistakes regarding the readability of the paper; for example in line 118 the word “be” should be added after “are going to”. The reference list needs to be checked again since I think that the paper in [1] refer to the paper published in JRSSB.

Reviewer 2



EDIT: I have read the response and the other reviews. I appreciate the clarification by the authors, and am content that they will include a more careful discussion of related work in the final version. Clarity: I found this paper generally clear and easy to read. I would suggest the following improvements for readability. - Include x in R(omega) and R to highlight the fact that it is a (random) function of x. - Using sans-serif to indicate when a quantity is random is nice, but its usefulness was largely obviated by the fact that the symbol for omega is serif vs. sans-serif is nearly identical. I would recommend either going with capital or using a symbol other than omega for the auxiliary variables. - I believe there is a type in Lemma 1 (should be Q(omega)R(omega)) Quality: The paper is generally of high quality and correct. Still, the current draft lacks a comprehensive related work. Especially in the context of questions about originality (below), this is a major omission. Originality: There is significant overlap between this work and known techniques in the Monte Carlo literature. The idea is very closely related to the idea of matching the proposal process to a distribution over an extended state space. This idea was used in Neal 2001, Andrieu et al. 2010, and reviewed in Axel Finke's thesis. Even more, I believe this idea is essentially equivalent to the idea proposed in Lawson et al. 2019 presented at an ICLR 2019 workshop. The focus of the Lawson et al. 2019 paper is different, in that they focus on the choice of the coupling. The Lawson et al. 2019 paper derives from work on auxiliary variable variational inference, see e.g. Agakov & Barber 2004. In any case, I urge the authors to spend more time explaining the connections between their work and the existing literature. Significance: This is an active research area, and this paper has the potential to generate interest. Even so, the experimental results are currently quite weak and some interesting avenues remain unexplored. - The improvements appear in Figure 6 to be on the order of a hundredth of a nat, which is a bit too small to justify changing a codebase to incorporate these bounds. - I would urge the authors to investigate how these distinct bounds interact with learning the parameters of q (the variational approximation) and p (the model). Citations: - Radford Neal. Annealed Importance Sampling. Statistics and Computing, 2001. - Axel Finke. On Extended State-Space Constructions for Monte Carlo Methods. - Christophe Andrieu, Arnaud Doucet, Roman Holenstein. Particle Markov chain Monte Carlo methods. JRSSB 2010. - Dieterich Lawson, George Tucker, Bo Dai, Rajesh Ranganath. Revisiting Auxiliary Latent Variables in Generative Models. ICLR Workshop 2019. - Agakov, Felix V., and David Barber. "An auxiliary variational method." International Conference on Neural Information Processing. Springer, Berlin, Heidelberg, 2004.

Reviewer 3



% Post rebuttal comments Thank you for your rebuttal. I am satisfied that the authors have taken on the feedback from the reviewers and are taking steps to improve the paper. I do still feel like there is a lot more that could be done with the experiments section in the paper beyond just the changes in the presentation proposed in the rebuttal and so I strongly encourage the authors to look into additional experiments for the camera-ready as well. I have decided to stick to my original score. %%%%% Overall I think this is a strong submission that would be of notable interest to the NeurIPS community. As explained above, I think the work has clear significance, while the approach also scores well on originality: although specific instances of the framework have already appeared in the literature (e.g. the appropriate Q(z) is known for IWAE and VSMC) and the objectives themselves are not really novel (i.e. the divide step is just prior work), there is clear novelty through the general nature of the coupling framework and the associated analysis which justifies it. The paper is clearly written and most of it is easy to follow (with the exception of the experiments section), despite the somewhat subtle nature of some of the ideas. The quality of the paper is also very good: I carefully checked the proofs for the theorems in the paper and found them to be correct, while a very detailed and precise measure-theoretic version of the results is also provided in the supplement. My main criticism with the paper is the experiments section, which feels like a bit of an after-thought. For example, the quality of the writing is far below that of the rest of the paper, there is a large unnecessary tangent about mappings of variance reduced sampling approaches in Gaussians (i.e. Figure 7 and accompanying text), and I do not feel it really provides an insightful investigation of the ideas being introduced. Though the results from Figure 8 are useful, they are both somewhat difficult to interpret and do not give the most emphatic of support for the suggested approaches. I feel there are a number of interesting empirical investigations that could have been made but were not, such as using the methods in a VAE context or doing more detailed investigations of the qualitative behavior of different Q(z). Other specific points 1. There appears to be a major typo in Lemma 1 and elsewhere: P(omega) is never defined, but is interpreted as Q(omega) it the vast majority of places (in the original version of this result from Le et al. 2018 [9] it is just Q(omega)). I presume this is just meant to be Q(omega) everywhere, but the fact that line 417 has P(omega)=Q(omega) makes me think the authors might be trying to convey something extra here. Please comment. 2. The KL is only one of many subjective metrics for the disparity between two distributions. This raises a number of interesting subtleties about whether Q(z) is an objectively better approximation, or simply one that is more suited to KL(Q||P) (e.g. its tendency to mode seeking). I would expect it to be the former, but I think the paper would be better for some discussion and maybe even experimentation on this. For example, if Q_0(omega) makes a mean field assumption, does Q(z) allow one to overcome this? 3. The definition of the KL for conditional distributions as being the expectation over the conditioning variable is liable to cause confusion as this is not standard in the VAE literature: it really threw me off the first time I read Theorem 2. I do not think anything is lost by writing these out explicitly (e.g. as an expectation of Q(z) in Theorem 2) and it would make it much things easier to interpret and avoid potential confusions like mine. 4. In lines 56-57 the paper talks about "novel objectives enabled by this framework". This is a bit of misnomer as the paper is not really contributing much in the way of new objectives: it is already known that the expectation of the log of any unnormalized marginal likelihood estimator is valid variational bound, which already implies all of these objectives which are standard marginal likelihood estimators. 5. I personally find it quite difficult to distinguish the sans-serif font for random variables. It might be good to think about a different way of distinguishing them. 6. The spacing in Table 1 should be fixed. 7. I would use something other than u in Figure 5 to avoid confusion with the "uniform" random variables \omega. 8. I really do not feel that Fig 7 and the accompanying text adds much to the paper. I would move it to the supplement and use the space to better explain the numerical experiments: the last paragraph of section 5 is extremely terse and it is really difficult to figure out exactly what experiments you are actually running (you almost have to read the Figure 8 caption before any of the main text to give things context). There is also no real discussion of the results which makes them difficult to interpret. In general, I think section 5 needs some pretty heavy editing. 9. I would explicitly define << on line 314 to make this more accessible as people may not be familiar with its typical measure theory meaning. 10. Line 334 has a tex error in the citation. 11. The proof block for Theorem 9 ends on line 380 instead of 387 for some reason. 12. In the proof of Theorem 2, I think you should spell out (i.e. add some words for) the change from P^{MC}(z|x) to p(z|x): this is somewhat the crux of the proof and it is left somewhat implicit as to why it is the case, some hand holding would help. The proof also seems to be a bit sloppy on jumping between have omega etc be scripted as random variables or not. 13. In the proof of Claim 10 the second delta should be \delta(z-T(\omega)) instead of \delta(z-\omega). The result is still the same.

[Author Response · NeurIPS 2019]

**[Related work]** We agree our work is related to MC methods on augmented spaces and will add a more discussion of
this to the paper. One distinction is that augmented MC methods traditionally use particle-based estimators where the
*choice* of coupling is "obvious", but the *proof* much less so (as we briefly allude on lines 41-44, 112 and 138-145), and
require case-by-case derivations if a new estimator is introduced. We start with an arbitrary estimator $R$, for which is is
not clear *a priori* that a coupling *exists*, and provide a systematic approach to finding estimator-coupling pairs.

In more detail, the closest MCMC work is "Particle Independent Metropolis Hastings" (Andrieu et al. JRSSB). It says (in
our terminology): take an estimator for $p(x)$ defined by $Q(\omega)$ and $R(\omega)$ that also comes with an "obvious" distribution
$a(z|\omega)$ for sampling $z$ (i.e., select one particle or trajectory in proportion to its weight). Define the extended proposal
distribution $Q(z, \omega) = Q(\omega)a(z|\omega)$ and the (unnormalized) extended target $P^{\mathrm{MC}}(z, \omega, x) = R(\omega)Q(\omega)a(z|\omega)$. Run
independent Metropolis-Hastings (MH) with extended target and proposal to sample from $P^{\mathrm{MC}}(z, \omega|x)$. The acceptance
probability is $\min(1, R(\omega')/R(\omega))$, so can be computed as simple byproduct of generating the proposal. Further, one
can show using properties of particle-based estimators that the "obvious" distribution $a(z|\omega)$ is indeed a coupling, i.e.,
$p(z|x)$ is a marginal of $P^{\mathrm{MC}}(z, \omega|x)$, so by discarding the $\omega$ variables we have a valid MCMC sampler for $p(z|x)$. The
underlying reasoning is the same as in our Theorem 2: $R$ is the ratio of the extended proposal and target densities used
within MH. Our work can be viewed as the "VI side" of this work — what happens if we use extended proposals and
targets within VI? In MCMC, dropping auxiliary variables automatically yields a valid marginal sampler. In VI, it
introduces the conditional divergence in the ELBO decomposition (Theorem 2).

R2 also mentions the recent ICLR workshop paper of Lawson et al. (which appeared in early May and is concurrent to
our work) and augmented VI more broadly. Our paper has strong roots in augmented VI and we certainly think of it as
such. We hint at this in a few places (lines 8, 11, 122, 156) this but can (and will) make the point more explicitly. We
build most closely on ref [7], which clearly articulates the special case of Theorem 2 for IWAEs (i.e., "IID Mean" in our
Table 2) as augmented VI. Lawson et al. arrive at a decomposition of $\log p(x)$ analogous to our Theorem 2 but from a
very different standpoint. They assert that "many unbiased estimators can be justified as performing simple importance
sampling on an extended state space", and assume knowledge of the relevant extended and conditional distributions (cf.
the form of $\hat{p}(x)$ at the end of Sec 2.2) — essentially a coupling in our terminology. They then *derive* the distributions
(only) for IWAEs, as was done in ref [7] (also [6], [14]). The details for other objectives are left unstated and require
case-by-case derivations. In contrast, we provide general tools to find estimator-coupling pairs without case-by-case
derivations. We also believe that our framework of estimator-coupling pairs much more explicitly and clearly articulates
the ingredients needed for such an approach to work.

**[Experiments]** We accept the view of the reviewers that our experiments did not make our points convincingly enough.
We will start by reformatting the presentation. Our primary point was that better likelihood bounds ($\mathbb{E} \log R$) correspond
to better posterior approximations, thus the two axes in Fig. 8. However, due to the large amount of model-to-model
variability in these two metrics, it is difficult to see the differences due to methods. We plan to move Fig. 8 to the
appendix and instead show results like Fig. 6 for more models — several examples are shown below (for likelihood
bounds). These show differences due to sampling strategies and numbers of replicates. We would like to emphasize that
iid sampling consistently improves on naive VI (equivalent to $M$=1) and that the alternative sampling methods offer a
further consistent improvement at near-zero computational cost. (Often, $M$=8 or $M$=16 with iid sampling is needed to
equal the performance of antithetic sampling with $M$=2.) We will also create analogous plots that aggregate across
models by normalizing/standardizing the two metrics (likelihood bound improvement, posterior error).

**[Writing]** The reviewers made many helpful points regarding the organization and writing of the paper, as well as
pointing out typos. We will revise the paper with these in mind, focusing especially on providing details and summary
of experimental findings, and de-emphasizing some of the current content in Sec. 5.



[Meta-Review · NeurIPS 2019]

An interesting and potentially influential contribution to the variational inference literature. The authors provides a framework for reinterpreting any Monte Carlo variational objective (e.g. based on IS or particle filtering) as an ELBO with a richer variational distribution, which means that the approach provides a new general way of enriching variational posteriors. The technique is demonstrated on several MC estimators of the likelihood based on variance reduction techniques. The experimental section is currently minimal and is more of a proof-of-concept, but given the significance of the conceptual and theoretical contributions of the paper this is not a significant issue. A conclusion section and the discussion of some missing connections to prior work pointed out by the reviewers, promised by the authors in their response will be very welcome additions to an already strong paper.